# Low Efficacy of *Isaria fumosorosea* against Box Tree Moth *Cydalima perspectalis*: Are Host Plant Phytochemicals Involved in Herbivore Defence against Fungal Pathogens?

**DOI:** 10.3390/jof6040342

**Published:** 2020-12-06

**Authors:** Rostislav Zemek, Jana Konopická, Zain Ul Abdin

**Affiliations:** 1Arthropod Ecology and Biological Control Research Group, Ton Duc Thang University, Ho Chi Minh City 758307, Vietnam; zainulabdin@tdtu.edu.vn; 2Faculty of Applied Sciences, Ton Duc Thang University, Ho Chi Minh City 758307, Vietnam; 3Biology Centre of the Czech Academy of Sciences, Institute of Entomology, 370 05 České Budějovice, Czech Republic; jkonopicka@seznam.cz; 4Department of Plant Production, Faculty of Agriculture, University of South Bohemia, 370 05 České Budějovice, Czech Republic; 5Department of Entomology, University of Agriculture, Faisalabad 38000, Pakistan

**Keywords:** *Buxus*, entomopathogenic fungi, invasive pests, virulence, alkaloids, antimicrobial activity

## Abstract

*Buxus* sp. is an important native and ornamental tree in Europe threatened by a serious invasive pest *Cydalima perspectalis*. The larvae of this moth are able to defoliate box trees and cause their death. The development of novel biopesticides targeting this pest might help protect *Buxus* trees grown wildly or in city parks. Laboratory experiments were conducted to assess the efficacy of entomopathogenic fungus *Isaria fumosorosea* strain CCM 8367 against *C. perspectalis*. The last-instar larvae of the box tree moth were treated by the suspension of fungus conidia at concentrations ranging from 1 × 10^4^ to 1 × 10^8^ spores per 1 mL. Fungus infection was observed mostly in pupae, but the maximum mortality did not exceed 60%, indicating a very low susceptibility of *C. perspectalis* to *I. fumosorosea*. Furthermore, a number of ungerminated fungal conidia were found on larval cuticles using a low-temperature scanning electron microscopy. Our data also reveal that the hydroalcoholic extract from *B. sempervirens* leaves significantly inhibits both the germination of *I. fumosorosea* conidia and fungus growth. It can be speculated that the strain CCM 8367 of *I. fumosorosea* is not a potent biocontrol agent against *C. perspectalis* and low virulence of the fungus might be due to the accumulation of host plant phytochemicals having antimicrobial activity in larval cuticle of the pest.

## 1. Introduction

Box trees, *Buxus* sp., are important evergreen shrubs occurring in natural *Buxus* forests [1] or grown as ornamental trees in city parks in Europe. They are now endangered by the box tree moth (BTM), *Cydalima perspectalis* (Walker) (Lepidoptera: Crambidae), which is a serious invasive pest native to Asia that was first detected in Germany in 2007 and has since invaded a large area causing significant damage [2,3]. This pest overwinters at the larval stage [4,5] and can have two to four generations per year in Europe depending on abiotic conditions [6]. Natural enemies do not suppress the *C. perspectalis* population, which is probably because this exotic species does not seem to be a good host for native parasitoids [4,7,8]. Thus, the pest is able to destroy *Buxus* tree completely in one season [9]. Some synthetic chemical insecticides are effective in *C. perspectalis* control [10]. Still, their application in natural habitats is problematic because of their adverse side effects on non-target species. Their frequent application possibly leads to the risk of resistance development in the pest.

The use of microbial biopesticides against BTM offers a unique alternate solution to broad-spectrum chemical insecticides. The best results have been obtained by using products based on *Bacillus thuringiensis* Berliner (Bacillales: Bacillaceae) while entomopathogenic nematode *Steinernema carpocapsae* (Weiser) (Rhabditida: Steinernematidae) was less successful [11]. Entomopathogenic fungi (EPFs) represent other promising biocontrol agents. Their advantages are that they do need to be ingested as they are able to penetrate the host cuticle and can be relatively easily produced [12]. A number of mycoinsecticides, most commonly based on *Beauveria bassiana* (Balsamo) Vuillemin (Hypocreales: Cordycipitaceae), *Metarhizium anisopliae* (Metsch.) Sorokin, (Hypocreales: Clavicipitaceae), *Isaria fumosorosea* (WIZE) Brown & Smith (Hypocreales: Cordycipitaceae), and *B. brongniartii* (Saccardo) (Hypocreales: Cordycipitaceae) have been developed in the world [13]. To our knowledge, only *B. bassiana* strain GY1-17 was tested against BTM in Korea, but larvae were not affected significantly [14].

The present study aimed to assess the possibility of fungal biocontrol of *C. perspectalis* by *I. fumosorosea*, which is known to be virulent to many insect species including a wide variety of butterflies and moths [15,16,17] and has received significant attention as a possible biological control agent for several economically important pests [18]. The obtained results showed low efficacy of the fungus against this pest. Therefore, additional experiments were conducted to test the hypothesis that the low effectiveness might be due to the antifungal activity of some host plant phytochemicals consumed by the moth larvae. Low-temperature scanning electron microscopy revealed that the number of *I. fumosorosea* conidia did not germinate. In vitro experiments confirmed that the hydroalcoholic extract of *Buxus* leaves suppressed spore germination and fungus growth.

## 2. Materials and Methods

### 2.1. Insects

Last-instar larvae of *C. perspectalis* were collected from unsprayed *Buxus sempervirens* trees located in a private garden in Staré Hodějovice (South Bohemia, Czech Republic, 49° N) and maintained in net cages at a room temperature with 16L:8D photoperiod for a few days before they were used in bioassays. Young twigs of untreated box trees collected in the vicinity of the Biology Centre, České Budějovice were provided as food and replaced with fresh ones when needed.

### 2.2. Entomopathogenic Fungus

*Isaria fumosorosea* strain CCM 8367 was used in this study. The strain was isolated from the pupa of the horse chestnut leaf miner, *Cameraria ohridella*, Deschka & Dimić (Lepidoptera: Gracillariidae) collected in the Czech Republic [19] and deposited in the Czech Collection of Microorganisms in Brno as a patent culture [20,21].

The fungus was grown on PDA medium (Sigma-Aldrich, Darmstadt, Germany) at 25 ± 1 °C and a 16L:8D photoperiod. After 10 days of incubation, the spore suspensions were prepared from each strain by scraping off conidia into the sterile solution of 0.05% (*v*/*v*) Tween^®^ 80 (Sigma-Aldrich, Darmstadt, Germany). The suspension was filtered through sterile gauze to separate the mycelium and clusters of spores. In uniform suspension, the spores were counted with a Neubauer improved counting chamber (Sigma-Aldrich, Darmstadt, Germany), and subsequently, the suspension was adjusted to the required concentration.

The viability of spores was verified using a standard germination test [22]. Ten drops from suspension were applied using a 1 μL inoculation loop on the surface of 2% water agar, which was poured in a thin layer onto the surface of a sterile slide. After the drops had dried, the slides were moved into a wet chamber and incubated at temperature 25 ± 1 °C for 24 h. The percentage of germinating spores was determined using an Olympus CH20 light microscope (Olympus Optical Co., Ltd., Tokyo, Japan); bright field, 400× magnification. The spore germination in all tests was 100%.

### 2.3. Bioassays

#### 2.3.1. The Efficacy of *I. fumosorosea* against *C. perspectalis*

Five concentrations of *I. fumosorosea* ranging from 1 × 10^4^ to 1 × 10^8^ spores per 1 mL were used in this experiment. The last-instar larvae of BTM in treated groups were individually immersed in the suspension of conidiospores of the fungus for five seconds (dip-test). All specimens in a control group were immersed in sterile solution of 0.05% Tween^®^ 80 only. Then, the larvae were placed into polystyrene Petri dishes (vented, inner diameter 90 mm, height 15 mm, Gosselin™, Borre, France) lined with moist filter paper (KA 0, Papírna Perštejn, Ltd., Perštejn, Czech Republic) and kept under constant conditions (25 ± 1 °C and 16L:8D photoperiod). Larvae were fed with *B. sempervirens* leaves, which were replaced daily until larva developed into pupa or died. The filter paper was also daily moistened by distilled water to maintain optimal humidity inside the Petri dishes. The insects were monitored daily for a period of three weeks to record insect development, mortality, and the development of mycosis on cadavers until all individuals died or adults emerged.

All bioassays described above were repeated twice; each replication tested 15 insect individuals. Mycosis on cadavers and emerged adults were documented by digital cameras Olympus SP-510 (Olympus Optical Co., Ltd., Tokyo, Japan) and Nikon Coolpix 4500 (Nikon Corporation, Tokyo, Japan) mounted on a tripod and using macro mode.

#### 2.3.2. Scanning Electron Miscroscopy of *I. fumosorosea* Conidia Germination on Cuticle of *C. perspectalis* Larvae

In vivo germination of fungal conidia on the insect cuticle was examined by low-temperature scanning electron microscopy (LT-SEM). BTM larvae were treated by immersing in suspension of *I. fumosorosea* conidia (concentration 5 × 10^7^ spores mL^−1^) and incubated for 0, 24, and 48 h at the temperature of 25 °C. The larvae were mounted on an aluminum stub using Tissue-Tek (C.C.T.D. Compound, The Netherlands). The samples were extremely fast (<10^−3^ K/s) frozen in vapor of liquid nitrogen. After freezing, the samples were transfered into a GATAN ALTO-2500 high vacuum cryo-preparation chamber (Gatan Inc., Abingdon, UK). The surface of the sample was sublimated (freeze-etched) for 5 min at the temperature of −95 °C and at −130 °C. After sublimation, the samples were sputter-coated with gold at the temperature of −130 °C. Coated samples were inserted into the chamber of a JEOL JSM-7401F Field Emission Scanning Electron Microscope (JEOL Ltd., Tokyo, Japan). Images were obtained by the secondary electron signal at an accelerating voltage of 4 kV and current 10 µA using an Everhart–Thornley Detector (ETD).

#### 2.3.3. The Effect of *B. sempervirens* Extract on *I. fumosorosea* Germination and Growth

Plant material was collected from untreated *B. sempervirens* trees grown in the Biology Centre garden. The extract used for the study was prepared at the concentration of 20% (*w/v*) by grinding 2 g of fresh leaves in 10 mL of solvent (water–ethanol 1:1 mixture). Analytic grade ethanol (Penta Ltd., Czech Republic) and distilled water were used. The mixture was filtered through filter paper (KA 0, Papírna Perštejn, Ltd., Czech Republic) to remove particulate materials, and one milliliter of fresh extract was spread on the surface of 2% water agar in Petri dish and left to dry for 24 h. Then, a suspension of *I. fumosorosea* conidia was applied using an inoculation loop on the surface. Germination was evaluated in 100 spores after 24 h of incubation at 25 ± 1 °C as described above. The control plate was treated with solvent only. The experiment was conducted in three replicates. Spore germination was documented by NIS-Elements Imaging Software and a Nikon Eclipse E200 microscope equipped with Nikon DS-Fi3 color camera (Nikon Corporation, Tokyo, Japan).

The effect *B. sempervirens* extract on fungus growth was measured by a modified inhibition zone assay [23]. A half mL of conidia suspension in 0.05% Tween^®^ 80 at a concentration 1 × 10^4^ spores mL^−1^ was spread evenly on potato dextrose agar (PDA) medium in a plastic Petri dish (diameter 90 mm). A hydroalcoholic extract of *B. sempervirens* leaves prepared as described above was applied on filter paper discs (diameter 14 mm) in a dose 150 µL per disc. Control discs were treated with 150 µL of the pure solvent. The solvent was allowed to evaporate, and paper discs were placed carefully in the center of PDA plates. After 7 days incubation at 25 °C, the plates were photographed by a digital camera Olympus SP-510 (Olympus Optical Co., Ltd., Tokyo, Japan) mounted on a tripod to document differences in fungus growth. Then, the area of plate in the center not covered by *I. fumosorosea* mycelium was measured using ImageJ, a Java-based image analysis software [24]. The assay was conducted using 10 dishes (replications) for both treatment and control.

### 2.4. Data Analysis

To analyze the effect of treatment on developmental time of *C. perspectalis* larvae and pupae, we fitted generalized linear models (GLM) with a Poisson error distribution and log link function. Mortality as well as germination data were analyzed using GLM with a binomial distribution and logit link. Treatment and replication were set as fixed effects. The analyses were performed in SAS^®^ Studio for Linux [25] using the GLM procedure (PROC GENMOD) of SAS/STAT^®^ module [26]. Means were separated by the least-square means (LSMEANS) statement of SAS with Tukey–Kramer adjustment for multiple comparisons. *p*-values <0.05 were considered statistically significant. Lethal concentrations (LC_50_ and LC_90_) were estimated using Probit analysis (PROC PROBIT). Differences in area not covered by *I. fumosorosea* mycelium were compared by an exact Wilcoxon two-sample test (PROC NPAR1WAY) of the SAS/STAT^®^ module.

## 3. Results and Discussion

Most BTM larvae successfully passed to pupa regardless of treatment (Table 1), and no statistically significant effect of treatment on the duration of the larval stage was observed (χ^2^ = 10.08, df = 5, *p* = 0.0730). Similarly, treatments had no significant effect on the duration of the pupal stage (χ^2^ = 0.19, df = 5, *p* = 0.9992), but higher mortality was observed in all treatments; the maximum mortality of 46.4% pupae was found with the highest concentration of fungal treatment.

Mycosis was observed only in treatments of 1 × 10^7^ and 1 × 10^8^ conidia per 1 mL when 10% and 28.6% of pupae cadavers, respectively, were obviously infected by the fungus (Table 1). Infection by *I. fumosorosea* was later confirmed when fungus sporulated (Figure 1).

Interestingly, several adults that emerged from larvae treated by the highest conidia concentration were malformed (Table 1, Figure 2) and died in 1–2 days. A similar effect was observed when *I. fumosorosea* was applied to *Spodoptera littoralis* (Boisd.) [27].

The cumulative mean mortality, including mortality in malformed adults, varied among treatments and reached a maximum value of 60% when larvae were treated by a suspension of 1 × 10^8^ conidia per 1 mL (Figure 3a). Thus, the highest mortality corrected for mortality in the control group [28] was only 42.9%. Although the effect of treatment on mortality was significant (χ^2^ = 18.67, df = 5, *p* = 0.0022) and no significant differences were found between replications (χ^2^ = 0.45, df = 1, *p* = 0.5004), the results indicate the very low susceptibility of *C. perspectalis* to *I. fumosorosea*.

The low efficacy is rather surprising, because the strain CCM 8367 of *I. fumosorosea* used in this study was previously found to be highly virulent against several pest species. For example, the mortality of pupae of *C. ohridella*, an invasive pest of *Aesculus hippocastanum* in Europe [29], treated by blastospores or conidia suspension of concentration 5 × 10^7^ spores per 1 mL reached 100% over a few days [20]. Later, the high efficacy of this strain was confirmed against *Spodoptera littoralis* (Boisd.) in which an application of CCM 8367 blastospores at a concentration of 5 × 10^7^ per 1 mL caused larval mortality >90% [27]. The high efficacy of CCM 8367 under laboratory conditions similar to that used in the present study was reported also against Colorado potato beetle, *Leptinotarsa decemlineata*, (Say) (Coleoptera: Chrysomelidae) larvae [30]. This indicated that the strain could be a prospective biocontrol agent, although some side effects against non-target natural enemies were also reported [31].

The log-probit regression line describing the relationship between concentration and mortality has a form *y* = −1.264 + 0.135*x* (Figure 3b), but the slope was not statistically significant (χ^2^ = 3.27, df = 1, *p* = 0.071). Thus, the extrapolated values of LC_50_ = 2.42 × 10^9^ and LC_90_ = 7.88 × 10^18^ were very high. For example, this contrasts with the LC_50_ and LC_90_ values of 1.03 × 10^6^ and 8.67 × 10^7^, respectively, reported for *L. decemlineata* [30].

LT-SEM imaging of *I. fumosorosea* conidia on the cuticle of BTM larvae revealed a high number of spores immediately after treatment (Figure 4a,b), but after 24 and 48 h of incubation, the number of attached spores seemed to be much lower, as we found them only in some places of larvae, usually as small groups or individual conidia. This indicates low conidial attachment to the larvae cuticle. Examination further showed that the number of spores did not germinate (Figure 4d–f). This finding might explain the low virulence of the fungus against *C. perspectalis* because the successful germination of fungus conidia on the host cuticle has been considered to be necessary for infection [32,33]. Several studies documented that the cuticle of some arthropods repress the germination of EPF spores or further development of germlings and appressoria formation [34,35]. One of the reasons might be the presence of antifungal compounds on the cuticle [36,37], which might be also case of *C. perspectalis*.

Results of in vitro experiments using *B. sempervirens* hydro-alcoholic extract revealed that this extract has a negative effect on the germination of *I. fumosorosea* conidia (Figure 5). In the control treated by solvent, the mean germination was 100% (SE = 0, *n* = 3), while on extract-treated agar, the mean germination was only 92.67% (SE = 0.88, *n* = 3) in average. The difference was statistically highly significant (χ^2^ = 31.36, df = 1, *p* < 0.0001).

The inhibition zone assay showed the suppression of *I. fumosorosea* mycelium growth on filter paper discs treated by *B. sempervirens* extract. The mean area not covered by mycelium was 0.02 ± 0.01 mm^2^ and 53.83 ± 19.86 mm^2^ in control and treated discs, respectively. The differences were statistically significant (Z = −2.184, *p* = 0.028).

Our findings indicate the presence of phytochemicals in box tree leaves having some activity against entomopathogenic fungi. Several secondary plant compounds were found to have a negative effect on the germination of *I. fumosorosea* blastospores, indicating that the presence of allelochemicals on a substrate (e.g., insect cuticle or leaf) may be an additional constraint to the survival of entomopathogenic fungi [38]. The *Buxus* trees contain a lot of alkaloids, some of which are sequestered by *C. perspectalis* larvae, while some are metabolized and/or excreted [39]. The antimicrobial activity of substances extracted from *B. sempervirens* by 65% ethanol were found earlier [40], and similar effects of box tree extracts were confirmed by other authors [41]. Thus, it is thus likely that BTM larvae use phytochemicals obtained from the host plant for their own defense against the invasion of microbial pathogens. This might explain the low efficacy of two strains of entomopathogenic fungi, *I. fumosorosea* (present study), and *B. bassiana* [14] against BTM.

It may be concluded that the strain CCM 8367 of *I. fumosorosea* is not a potent biocontrol agent against *C. perspectalis* and that the reason for the low efficacy of the fungus might be the accumulation of host plant phytochemicals with antimicrobial activity in the fifth-instar larvae cuticle of the pest.

## Figures and Tables

**Figure 1 jof-06-00342-f001:**
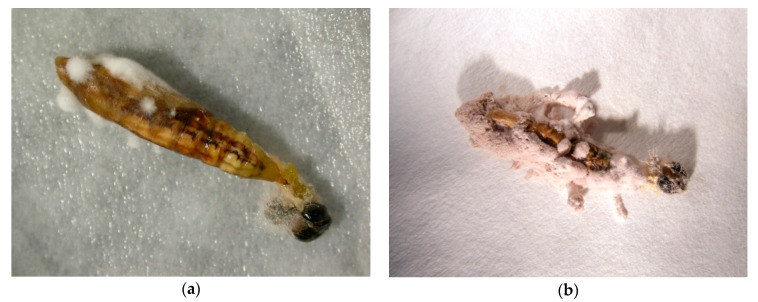
(**a**) Early mycosis of *Isaria fumosorosea* on *Cydalima perspectalis* pupa; (**b**) Cadaver of *C. perspectalis* covered with sporulating fungus.

**Figure 2 jof-06-00342-f002:**
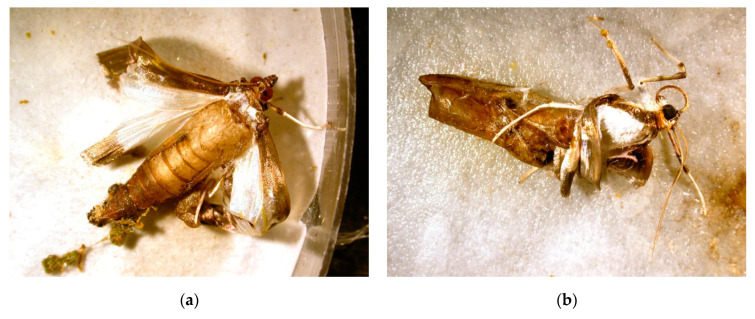
(**a**,**b**) Malformed adults of *Cydalima perspectalis* emerged in a group of larvae treated by *Isaria fumosorosea* at a concentration 1 × 10^8^ conidia/mL.

**Figure 3 jof-06-00342-f003:**
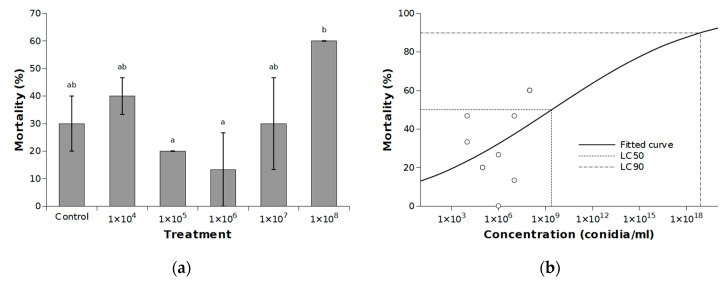
(**a**) Mean cumulative mortality (±SE) of *Cydalima perspectalis* (including mortality of malformed adults) treated with various concentrations of *Isaria fumosorosea* conidia. A generalized linear model was fitted and pairwise between treatment differences were tested using the least-square means. Different letters indicate significant differences between columns (*p* < 0.05); (**b**) Log-probit regression line of concentration-mortality response of *C. perspectalis* to *I. fumosorosea*.

**Figure 4 jof-06-00342-f004:**
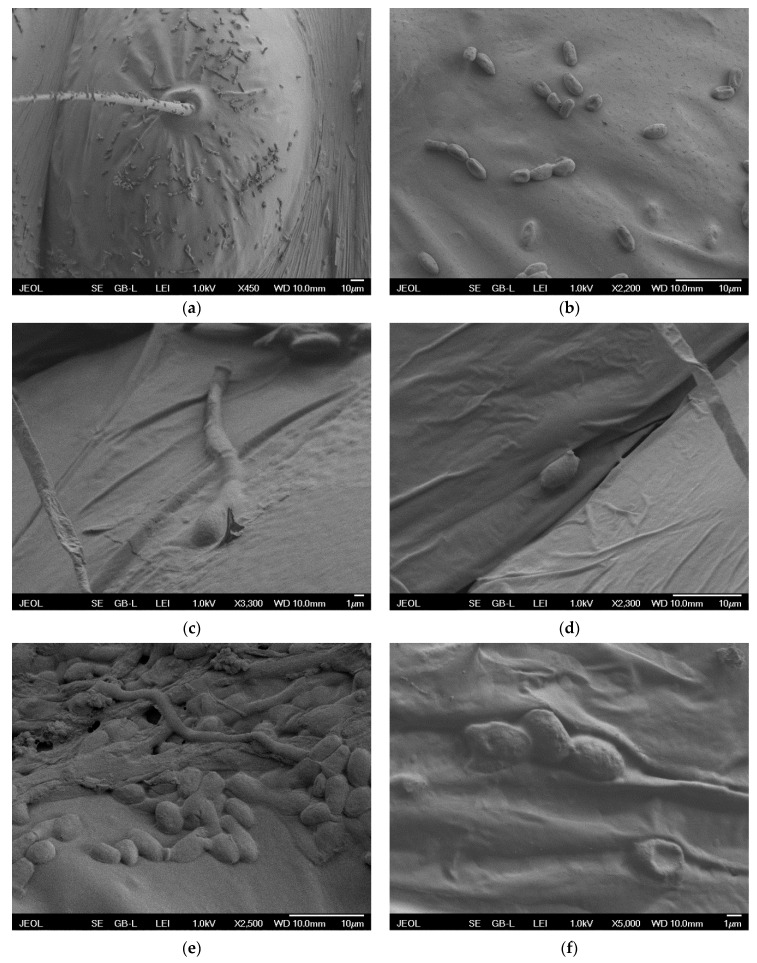
Low-temperature scanning electron microscope (LT-SEM) image of *Isaria fumosorosea* conidia on the cuticle of *Cydalima perspectalis* larva. (**a**,**b**) Conidia immediately after the fungus application; (**c**) Conidium with germ tube after 24-h incubation; (**d**) Ungerminated conidium after 24-h incubation; (**e**) Group of ungerminated conidia after 24-h incubation; (**f**) Ungerminated conidia after 48-h incubation.

**Figure 5 jof-06-00342-f005:**
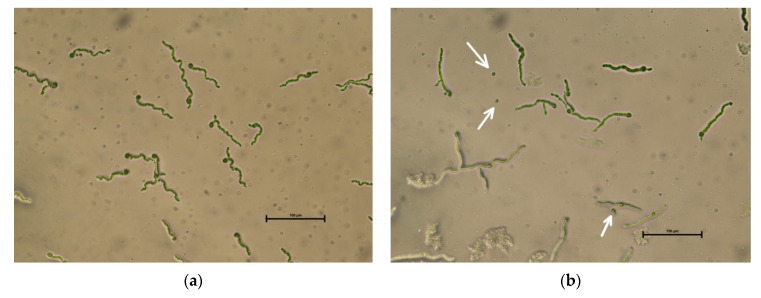
Germination of *Isaria fumosorosea* conidia on: (**a**) a control agar plate treated with solvent only; (**b**) agar plate treated with *Buxus sempervirens* extract. Arrows indicate spores with no or little germination peg. Traces of plant extract are visible on the agar surface.

**Table 1 jof-06-00342-t001:** The effects of *Isaria fumosorosea* on the development of *Cydalima perspectalis*.

Treatment ^1^	Last-Instar Larvae			Pupae			Malformed Adults
	Duration		Died/Mycosed	Duration		Died/Mycosed	
	Mean ± SE	*n*	%	Mean ± SE	*n*	%	%
Control	4.24 ± 0.41	29	3.3/0	10.10 ± 0.15	21	27.6/0	0
1 × 10^4^	5.04 ± 0.58	28	6.7/0	10.33 ± 0.11	18	35.7/0	0
1 × 10^5^	5.53 ± 0.54	30	0/0	10.17 ± 0.13	24	20.0/0	0
1 × 10^6^	4.87 ± 0.43	30	0/0	10.46 ± 0.10	26	13.3/0	0
1 × 10^7^	5.00 ± 0.31	30	0/0	10.33 ± 0.11	21	30.0/10.0	0
1 × 10^8^	3.96 ± 0.32	28	6.7/3.3	10.27 ± 0.25	15	46.4/28.6	20.0

^1^ Concentration of conidia per milliliter of suspension.

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
