# Peer review of "Low Efficacy of Isaria fumosorosea against Box Tree Moth Cydalima perspectalis: Are Host Plant Phytochemicals Involved in Herbivore Defence against Fungal Pathogens?"

_jof, 2020, doi:10.3390/jof6040342_

Round 1

Reviewer 1 Report

The authors have followed the recommendations and have addressed all suggested additional trials. I consider that the communication is now publishable after address the minor corrections that are detailed below:

- Most importantly, please clarify throughout the manuscript that the LC50 was extrapolated but not estimated (line 24 and so on), since the Probit test carried out does not meet the conditions about wide range of mortalities obtained. Even better if limit the sentences to “very low susceptibility” without numerical values when possibly.

- Line 100: Please clarify how many seconds were insects submerged in the “dip-test”.

- Line 123: Fungal conidia.

- Line 136: The effect of B. sempervirens extract on I. fumosorosea germination and growth.

-  Line 149: Do not start a sentence with a number, please spell it.

- Figure 4 legend: (e) is missing and (f) is repeated.

- Line 232: delete “only”.

Author Response

We would like to thank for all comments and suggestions which we all incorporated in revised version as specified below.

The authors have followed the recommendations and have addressed all suggested additional trials. I consider that the communication is now publishable after address the minor corrections that are detailed below:

- Most importantly, please clarify throughout the manuscript that the LC50 was extrapolated but not estimated (line 24 and so on), since the Probit test carried out does not meet the conditions about wide range of mortalities obtained. Even better if limit the sentences to “very low susceptibility” without numerical values when possibly.

Response: Sentence on LC50 value (line 24-25) was deleted in Abstract and verb “estimated” replaced with “extrapolated” in Results (now line 208).

- Line 100: Please clarify how many seconds were insects submerged in the “dip-test”.

Response: Text was modified to specify the time: “… were individually immersed in the suspension of conidiospores of the fungus for five seconds (dip-test).”

- Line 123: Fungal conidia.

Response: Text was corrected.

- Line 136: The effect of B. sempervirens extract on I. fumosorosea germination and growth.

Response: The section title was modified as suggested.

-  Line 149: Do not start a sentence with a number, please spell it.

Response: The 0.5 replaced by “Half”.

- Figure 4 legend: (e) is missing and (f) is repeated.

Response: This typo was corrected.

- Line 232: delete “only”.

Response: Word “only” was deleted.

Reviewer 2 Report

The resubmitted version of "Low efficacy of Isaria fumosorosea against box tree moth Cydalima perspectalis" deals with the comments of both reviewers.

The authors extended the Materials and Methods and other parts of the MS.

The new agar plate assays did not show meaningul inhibition of fungal growth, merely a lack of colonization on the paper disks was observed in 70% of the assays. The electron microscopic images provide some additional information to the observations, although by nature they can not be statistically evaluated in this case.

With the corrections and the few new additional data, this work can be considered as a short comm for publication, even though it certainly has its limitations. But as the authors pointed out, the original idea was indeed a short comm, to call attention on the potential effect of the host plant extract on an otherwise potent insect pathogenic fungus.

Some new typos were included!!! Consider this:  hydroalcoholic extract of Buxus leaves suppress spor germination and fungus growth. Or: emmerged.

Author Response

We would like to thank for all comments and suggestions which we all incorporated in revised version as specified below.

The resubmitted version of "Low efficacy of Isaria fumosorosea against box tree moth Cydalima perspectalis" deals with the comments of both reviewers.

The authors extended the Materials and Methods and other parts of the MS.

The new agar plate assays did not show meaningul inhibition of fungal growth, merely a lack of colonization on the paper disks was observed in 70% of the assays. The electron microscopic images provide some additional information to the observations, although by nature they can not be statistically evaluated in this case.

Response: Revised version now includes area measurements by image analysis of agar plate photos in growth-inhibition assay and statistical comparison of these data.

With the corrections and the few new additional data, this work can be considered as a short comm for publication, even though it certainly has its limitations. But as the authors pointed out, the original idea was indeed a short comm, to call attention on the potential effect of the host plant extract on an otherwise potent insect pathogenic fungus.

Response: We agree that results presented in the manuscript are limited but we believe that even data are preliminary they show interesting interaction and are worth to be published as Commnunication.

Some new typos were included!!! Consider this:  hydroalcoholic extract of Buxus leaves suppress spor germination and fungus growth. Or: emmerged.

Response: These typos were corrected.

Reviewer 3 Report

Dear Authors, please remove figure 2 and figure 6.

Why 10^4 have more mortality than 10^5, 10^6 and 10^7 (Table 1)? You dont discuss your result and Your title is Low efficacy of Isaria fumosorosea against box tree  moth Cydalima perspectalis: Are host plant  phytochemicals involved in herbivore defence  against fungal pathogens?

I suggest to remove the section 2.3.3. The effect of B. sempervirens extract on germination of I. fumosorosea conidia and fungus grow. Also i suggest a new title Efficacy of Isaria fumosorosea against box tree moth Cydalima perspectalis.

Author Response

We would like to thank for all comments most of which we incorporated in revised manuscript.

Dear Authors, please remove figure 2 and figure 6.

Response: We removed photos c and d from Fig. 2 as they showed healthy adults which were indeed redundant but we would like to keep photos a and b as they demonstrate the malformations. Fig. 6 was removed and replaced with results of image-analysis. Its results and statistics are now presented in text at lines 236-241.

Why 10^4 have more mortality than 10^5, 10^6 and 10^7 (Table 1)? You dont discuss your result and Your title is Low efficacy of Isaria fumosorosea against box tree  moth Cydalima perspectalis: Are host plant  phytochemicals involved in herbivore defence  against fungal pathogens?

Response: Text of paragraph starting at line 183 was rewritten and corrected mortality (using Abbott equation) was added to emphasize low efficacy. We agree that it is strange that higher mortality was found at lower concentration nevertheless there were no significant differences between 10^4 and other treatments, it is due to high variability in data. Since mycosis was found only at concentrations 10^7 and 10^8 the mortality at lower concentrations are probably caused by other causes than pathogen.

I suggest to remove the section 2.3.3. The effect of B. sempervirens extract on germination of I. fumosorosea conidia and fungus grow. Also i suggest a new title Efficacy of Isaria fumosorosea against box tree moth Cydalima perspectalis.

Response: We would prefer to keep this unchanged because by removing part on antifungal effect of host-plant extract the paper would loose important message on possible indirect interaction between host plant and entomopathogen. We are aware that our paper presents preliminary data (therefore it is “Communication”) but it can provide inspiration for other research revealing why this invasive pest is so successful.

Round 2

Reviewer 3 Report

Dear Authors,

it is ok for me 

This manuscript is a resubmission of an earlier submission. The following is a list of the peer review reports and author responses from that submission.

Round 1

Reviewer 1 Report

Dear Authors,

after reviewing the ms with Low efficacy of Isaria fumosorosea against box tree moth Cydalima perspectalis: Are host plant phytochemicals involved in herbivore defence against fungal pathogens?. I believe is very good work but isnt ready yet for an article. I believe is better as short communication.

Why short communication because you don’t do a full instar evaluation (first, second, three, fourth, five) and the pupae of the insect.  The second is that the M&M is not your better section because is not written good. You must explain how the bioassay is done. Personally I don’t understand what you do. Third the C. perspectalis rearing is poor, please add some information.

The Results and Discussion section, the major fault is between lines 146-150 (figure 3). The meaning of that figure for me is that I. fumosorosea strain had no effect on the mortality o the larvae. Why do you present this data? Control and the treatment is the same in terms of the statistics. Im certain that the experiment you made wasn’t so good.

In conclusion I must suggest more experiments for article and for communication must you done very good work to rewrite some section of the ms

Reviewer 2 Report

The manuscript „Low efficacy of Isaria fumosorosea against box tree moth Cydalima perspectalis: Are host plant phytochemicals involved in herbivore defence against fungal pathogens?” was submitted to JoF by Zemek et al. The Communication manuscript is intended to be included in the Host-Pathogen Interactions: Insects vs Fungi issue, and as the title suggests, the topic is well suited.

The results are mostly negative in a sense that the fungus proved to be a rather ineffective agent against the box tree moth, but the authors demonstrated a possible interesting explanation to this phenomenon. The study design is sound, there is an appropriate number of citations.

My minor comments:

At the first mention of Isaria fumosorosea, add taxonomic information

Their frequent application leads to the risk of resistance development in the pest: is this based on observations with related species? Any references? If not, rephrase as a possibility.

Bacillus thuringiensis while entomopathogenic nematode Steinernema carpocapsae (Rhabditida: Steinernematidae): you add taxonomic information for the animal, but not the bacterium., You gave BTM a full taxonomic and author + year description, but not for the nematode. These things should be consistent.

Line 94: what type of ventilation? Ventilated dishes? How high?

Figure 2: maybe consider adjusting the white balance using any photo editor software.

Typos: fungus again this pest

the horse chestnut, Cameraria ohridella: the horse chestnut is a plant!

2.3.1. The efficacy of I. fumosorosea against C. perspectalis; 2.3.2. The effect of B. sempervirens extract on germination of I. fumosorosea conidia: use italics where needed!

Reviewer 3 Report

The communication by Zemek et al. is devoted first to study pathogenicity of the entomopathogenic fungus Isaria fumosorosea strain CCM 8367 against larvae of the box tree moth, Cydalima perspectalis, under laboratory conditions. Then, trying to explain the low efficacy of the fungus towards this invasive pest, authors addressed the effect of hydroalcoholic extract from Buxus sempervirens leaves on fungal germination, in order to hypothesized that accumulation of host plant phytochemicals posse antimicrobial activity on insect cuticle. However, this is a speculation since no assays of germination on cuticle were performed.

The manuscript is in general well written. I have some concerns about methodology:

  1. Ideally, fungal concentrations used for Probit analysis must range between those with very low to those causing over total mortality in insect host. In this case, mortalities are low even at the higher dose used. Concentration of about 1×109 and 1×1010 conidia/ml should be used. The extrapolation of these data to obtain a LD50 is not realistic.
  2. Concerning the germination assay, although statistically significant, the difference between 100% (control) and 92.7% (treatment) is not too important in the (absolute) quantitative point of view. Moreover, the pictures selected seem to be little impact to show this point, they should be taken out of the manuscript (the same as some pictures of Fig. 1, only one photo from panel a and b and one from from panels c and d are enough). I recommend to address an “inhibition zone assay” by embedding paper dishes with the hydroalcoholic extract to demonstrate that fungal growth is also inhibited by the plant extract when spreading through the agar (for methodology example, see Chemoecology 11:225–229, 2001).

Minor comment:

Line 68: Is the larval stage used known? Is it possible to obtain a laboratory colony from this insect? If not, it should be indicated. In line 92 is written that bioassays were conducted with last-instar larvae. This stage was field collected or larvae were rearing until last stage in laboratory?